# Sleep architecture and quality and pain experience in individuals with persistent low back pain and asymptomatic controls

Xuewen Wang[1]*, Jennifer M. C. Vendemia[2], Erin E. Kishman[1¤a], John R. Gilliam[1¤b], Alexandria M. Reynolds[2], Sheri P. Silfies[1]

**1** Department of Exercise Science, University of South Carolina School of Public Health, Columbia, South Carolina, United States of America, **2** Department of Psychology, University of South Carolina College of Arts and Sciences, Columbia, South Carolina, United States of America

¤a Current address: Department of Health and Exercise Science, College of Health and Human Sciences, Colorado State University, Fort Collins, CO USA
¤b Current address: Laboratory for Rehabilitation Neuroscience, University of Florida, Gainesville, FL USA
* xwang@sc.edu

## Abstract

### Introduction

Both pain and sleep have a broad impact on health and well-being. There is a bi-directional association between pain and sleep, but how sleep and pain are associated chronically remains unclear. This study examined the associations of sleep architecture and quality in relation to pain experience and its impact in individuals with persistent low back pain (pLBP) and asymptomatic controls.

### Methods

Participants included individuals in a current episode of low back pain with symptoms impacting function and persisting greater than three months (pLBP group, n = 20) and asymptomatic controls (control group, n = 19). A home sleep test device (Zmachine® Insight + , General Sleep Corporation) was used for three nights to ecologically assess sleep architecture and quality. Pain, psychosocial factors, and lumbar movement control were evaluated using standard testing.

### Results

The participants were 25.0 ± 4.8 yrs (mean±SD) with similar height, weight, and waist and hip circumferences in the pLBP and control groups. Deep sleep time was shorter (p = 0.034) for the pLBP (1.4 ± 0.4 hr) compared to the control group (1.7 ± 0.3 hr). For the pLBP group, deep sleep time and some sleep quality measures were associated with several pain-related anxiety, daily function, and impact measures independent of total sleep time.

**Data availability statement:** Data are held at OSF, an open platform, https://doi.org/10.17605/OSF.IO/VURCK

**Funding:** National Institutes of Health under Award Number R01HD095959 to University of South Carolina, PI: Vendemia and Silfies. The University of South Carolina ASPIRE 80004373, PI: Wang.

**Competing interests:** The authors have declared that no competing interests exist.

## Conclusion

These results indicate associations of sleep architecture and quality with pain experience in individuals with pLBP. Of all the sleep stages, the deep sleep stage may be more impacted by chronic pain than other stages.

## Introduction

Pain is a distressing experience associated with actual or potential tissue damage involving sensory, emotional, cognitive, and social components [1]. Pain may cause sensorimotor integration in the brain due to anticipation and unpleasant affective manifestation and result in maladaptive changes in biomechanical and movement patterns [2,3]. Approximately 20% of US adults experience chronic pain, and 8% have high-impact pain [4]. Low back pain, in particular, is a leading cause of disability and is responsible for 70.2 million years lived with disability globally in 2021 [5]. The management of low back pain remains challenging despite advancements in the understanding of its underlying mechanisms, risk factors, and treatment methods.

Approximately 72.1% of individuals with chronic back pain experience poor sleep quality and 68.9% have insomnia [6]. Sleep is a complex physiological and behavioral process essential for maintaining psychological well-being and optimizing function across multiple physiological systems. Sleep cycles through stages of light, deep, and rapid eye movement (REM) sleep. Each stage is characterized by different physiological, behavioral, neurochemical, and electrophysical attributes [7], and researchers believe that different stages of sleep may be responsible for unique physiological and psychological processes. Research recognizes a bidirectional association between pain and sleep. Pain potentially interferes with sleep by increasing the internal arousal level [8]. Conversely, disrupted sleep can increase pain sensitivity and enhance pain perception through multiple pathways [9]. However, due to the interrelationships among sleep stages and parameters, it is critical to examine sleep architecture and quality in relation to pain in individuals with chronic pain to improve our understanding of the accumulated impact and association between sleep and chronic pain.

Subjective assessment of sleep has commonly been used in previous research examining associations between sleep and pain; however, subjective assessment may be prone to bias and does not provide information on the architecture of sleep. Of the few studies that have examined the architecture of sleep, most have focused on fibromyalgia. A meta-analysis of these studies reported that individuals with fibromyalgia experience less sleep, poorer sleep efficiency, longer wake time after sleep onset and more light sleep [10]. Studies conducted in other chronic pain groups (e.g., rheumatoid arthritis, osteoarthritis, headache, and temporomandibular pain) [11] suggests that the specific pain condition influences the dynamics of sleep and pain. A systematic review of sleep and chronic spinal pain (e.g., neck, back) reported a weak to moderate association between self-reported sleep parameters and pain

[12]. However, the methodological quality of included studies was low to moderate and the lack of wearable sleep tracking make it unclear how sleep architecture and quality, particularly when ecologically assessed, associate with chronic or persistent low back pain (pLBP) [12].

Psychosocial measures, including pain-related fear of movement, anxiety, and depression, are recognized as significant factors in the development of pLBP [13,14]. Persistent pain and movement impairment can also be explained by models involving central pain processing changes, individuals' pain experience, and their beliefs about pain [2]. Examining sleep architecture and quality in relation to the various psychosocial aspects of pain experience and the impact of pain will improve our understanding of the sleep and pain associations.

Therefore, the purpose of this study was to improve our understanding of the sleep and pain association by ecologically assessing sleep using an ambulatory electroencephalography (EEG) sleep monitor which allows the determination of sleep architecture and quality over several nights at home, and to assess the impact of pain on psychosocial well-being and movement control in individuals with pLBP and asymptomatic controls.

## Methods

### Participants

This study was an ancillary study to the "Linking Altered Cortical Sensorimotor Integration with Movement Impairments in Low Back Pain" study, in which participants' clinical assessments of pain, lumbar movement control, and related psychosocial factors were evaluated. Participants were provided with information about this study and asked to indicate interest in participating during the initial session of the parent study. Those who were interested were provided with detailed information and invited to participate. The University of South Carolina Institutional Review Board (Pro00112233) approved the study protocol. The privacy rights of human subjects were observed, and all participants signed a written informed consent form before participating. Recruitment of participants for this ancillary study occurred between July 3, 2021 to March 31, 2023.

The participants in the parent study included adults between the ages of 18–65 years with pLBP and sex and age-matched (±5 years) asymptomatic controls. Individuals with any conditions or who used medications that could influence any primary outcomes of the parent study (e.g., clinical levels of depression, anxiety, and other psychiatric disorders) were excluded.

Inclusion criteria for pLBP were: (1) current low back pain episode with symptoms persisting greater than three months, and (2) back pain impacting function during a minimum of half the days in the week during the last 3–6 months (U.S. National Institutes of Health Chronic Low Back Pain Minimal Dataset (NIH Dataset)) [15]. The inclusion criteria intentionally sought to sample the subcategory of high-impact chronic pain based on subjects' perception of how much pain limited daily routines. Asymptomatic controls included those who had not sought treatment for low back pain and had not lost the ability to accomplish daily tasks for more than three days during the last five years due to low back pain.

This ancillary study did not have additional inclusion or exclusion criteria. Due to the timing of this ancillary study in relation to the parent study and additional consent, six participants could not be sex and age matched (±5 years) and were excluded from the final analysis. Excluded participants (female: pLBP=2, control=1; male: pLBP=2, control=1) were older than the included participants (mean: 54.0 vs. 25.0 years).

### Measurements

After enrollment, anthropometric measurements were collected during the first visit and a home sleep test device (Zmachine® Insight+, General Sleep Corporation, Cleveland, OH) was provided with instructions for use at home over three nights. During the following visit, participants returned the device and were asked to complete several questionnaires.

**Objective sleep assessment.** The Zmachine® Insight+ is an FDA-approved device that uses state-of-the-art EEG hardware and advanced signal processing algorithms to detect sleep stages. A previous study found sensitivities and

positive predictive values for sleep stages using the algorithms, compared to the in-lab polysomnography, were between 0.72 and 0.85 and the overall kappa statistics was 0.72, indicating substantial agreement between the two methods [16]. The device acquires the EEG signal using three self-stick, single-use, disposable sensors that are self-applied by the participant on the back of the ears and the neck. Participants were instructed to use the device for three nights, generally considered necessary to obtain mean values to represent habitual sleep. They were instructed to avoid nights that they knew were atypical, for example, going to bed later due to a scheduled event. The manufacturer-provided software (Zmachine Data Viewer Version 3.5.1.0) was used to analyze data. Parameters are calibrated to an individual's EEG patterns. Obtained variables include sleep period time (total recording time from trying to sleep to stop recording), total sleep time (TST), duration of sleep stages (light sleep, deep sleep, and REM sleep) and their percentages of TST, as well as sleep quality measures (sleep efficiency, sleep latency, and wake after sleep onset). The mean values of the three nights were calculated.

**Subjective sleep assessment.** The risk of sleep apnea was assessed by the Berlin Questionnaire which classifies the overall risk as high or low risk [17]. The NIH Dataset [15] participants completed as part of the parent study assessments contained four sleep-related questions: sleep quality, refreshing, problems, and difficulties over the previous seven days. The answers ranged from "not at all" to "very much" (scores 1–5). Questions were reverse-coded as needed so that higher scores indicated better sleep.

**Pain and psychosocial assessments.** Participants in the pLBP group completed several assessments of their pain experience in the parent study. The NIH Dataset assesses the impact of pain defined by pain intensity, interference, and impact on physical function [15]. The Chronic Pain Acceptance Questionnaire (CPAQ) measures acceptance of pain which includes two subscales: activity engagement (pursuit of life activities regardless of pain) and willingness to experience pain (recognition that avoidance and control are often unworkable methods of adapting) [18]. The Tampa Scale for Kinesiophobia assesses fear of physical movement and activity resulting from a feeling of vulnerability to painful injury or re-injury [19]. It has two constructs: activity avoidance and somatic focus. The Pain Anxiety Symptom Scale (PASS) measures fear and anxiety responses specific to pain [20]. It is comprised of four components: cognitive, fear, escape/avoidance, and physiological. The Oswestry Disability Index assesses how back pain affects an individual's ability to manage everyday life [21].

For the entire sample, the Pain Catastrophizing Scale assessed pain experience, it measures how individuals feel and what they think about when they are in pain [22]. Depressive symptoms were assessed by the Center for Epidemiological Studies-Depression (CES-D) Scale [23]. The State-Trait Anxiety Inventory measured trait and state anxiety, which distinguish anxiety from depressive syndromes [24]. In addition, habitual physical activity was assessed using the Baecke Physical Activity Questionnaire [25]. Participants also reported their race/ethnicity, sex, and education level.

**Lumbar movement control evaluation.** A battery of six tests was conducted to evaluate movement control of the lumbar spine following the protocol by Luomajoki et al, which had substantial intra and inter-rater reliability (kappa >0.60) in both novice and experienced clinicians [26,27]. The tests included waiter's bow, pelvic tilting, quadruped rocking backward and forward, single-leg standing, seated knee extension, and prone knee flexion. Except for pelvic tilting, a test was considered positive when the subject could not stabilize the lumbar spine and prevent undesired low back movement during the task. The total number of positive tests was calculated (maximum 6 points, with higher scores indicating less control of the low back region). The number of positive tests was significantly different between participants with versus without low back pain in a previous study [26].

## Statistical analysis

This study was ancillary to the parent study; therefore, the sample size was primarily determined given the availability of resources and the feasibility of adding this component to the parent study. It was estimated that a sample size of 25 in each group would provide 80% power to detect a large effect size (Cohen's d = 0.81) for comparing means between the

two groups, using a significance level of 0.05 with a two-tailed t test. Initial analyses included descriptive analyses and examination of data distributions. Comparisons between pLBP and control groups were conducted by independent t-tests for continuous variables or chi-square tests for categorical variables. Comparisons of sleep variables with adjustment of covariates were made by analysis of covariance (ANCOVA) tests. Bivariate partial correlations of sleep variables with pain experience, psychosocial factors, and lumbar movement control were examined using Pearson's or Spearman's correlations. Age and sex were adjusted in the first model (model 1). TST was additionally adjusted in the subsequent model (model 2). Analyses were conducted using SAS version 9.4 (SAS Institute, Cary, NC). Statistical significance was defined as $p < 0.05$.

## Results

### Participant characteristics

The participants were young adults, 72% female, and 43.6% normal weight. As shown in Table 1, participants in the pLBP group and the control group had similar age, proportion of sex, racial/ethnic distribution, education, height, weight, body mass index, waist and hip circumferences, and physical activity. The two groups differed in several psychosocial measures. Compared to the control group, the pLBP group had greater depressive symptoms (CES-D), tendency to catastrophize pain, and state and trait anxiety although no measure met clinical diagnostic criteria. The pLBP group exhibited poorer lumbar movement control, as indicated by more positive results on the lumbar movement control test battery.

**Table 1. Participant characteristics.**

|  | pLBP<br>N = 20 | Control<br>N = 19 | *P* value |
|---|---|---|---|
| Age, years | 24.8 ± 5.6 | 25.2 ± 3.9 | 0.796 |
| Female, n (%) | 14 (73.7%) | 14 (70.0%) | 0.798 |
| Race/ethnicity |  |  | 0.163 |
| White, non-Hispanic | 12 (60%) | 17 (89.5%) |  |
| Black, non-Hispanic | 2 (10%) | 0 |  |
| Asian | 5 (25%) | 2 (10.5%) |  |
| Hispanic | 1 (5%) | 0 |  |
| Education |  |  | 0.514 |
| Some college, n (%) | 7 (35.0%) | 3 (15.8%) |  |
| Bachelors, n (%) | 7 (35.0%) | 9 (47.4%) |  |
| Graduate, n (%) | 6 (30.0%) | 7 (36.8%) |  |
| Height, m | 168.9 ± 8.4 | 168.6 ± 8.6 | 0.917 |
| Weight, kg | 78.1 ± 17.5 | 71.5 ± 12.9 | 0.187 |
| Body mass index, kg/m$^2$ | 27.3 ± 5.2 | 25.1 ± 3.4 | 0.121 |
| Waist circumference, cm | 85.9 ± 12.8 | 79.5 ± 9.2 | 0.086 |
| Hip circumference, cm | 105.4 ± 10.1 | 101.0 ± 7.2 | 0.128 |
| Beacke Physical Activity Questionnaire (range: 3–15) | 9.3 ± 1.8 | 9.8 ± 1.4 | 0.329 |
| CES-D score (range: 0–60) | 13.4 ± 9.0 | 6.3 ± 6.5 | 0.009 |
| State anxiety (range: 20–80) | 45.9 ± 5.2 | 42.3 ± 3.3 | 0.015 |
| Trait anxiety (range: 20–80) | 38.4 ± 11.0 | 31.1 ± 7.1 | 0.018 |
| Pain Catastrophizing Scale (range: 0–52) | 9.7 ± 8.1 | 3.2 ± 3.9 | 0.003 |
| Movement control tests, # positive, (range: 0–6) | 3.4 ± 1.0 | 0.9 ± 0.9 | < 0.001 |

Mean ± SD or number (percentage) are reported. P values are for comparison between groups from independent t-test or Chi-square test. CES-D, Center for Epidemiological Studies-Depression scale.

## Pain characteristics of participants with pLBP

Of the participants in the pLBP group, 5% (1 of 20) reported ongoing pain for three months, 30% (6 of 20) for six months to one year, 35% (7 of 20) for one to five years, and 30% (6 of 20) for more than five years. As to the frequency of pain, 50% reported every day or nearly every day and 50% reported pain on at least half of the days. Lower extremity activities were not limited by pain. Interventions currently used to control pain included: One participant used prescribed opioids, two used received an epidural or facet injection, 11 used exercise therapy, and one used psychological counseling. Means, SD, and quartiles of questionnaire scores for evaluation of pain experience are presented in Table 2. The average intensity of low back pain was 4.6 (±1.6, SD), where a rating of 1 was no pain, and 10 was the worst imaginable pain in the past seven days. The Oswestry Disability Index score indicated that most participants with pLBP reported mild disability due to back pain during specific activities of daily living (e.g., walking, sitting, standing, and personal care).

## Comparison of sleep characteristics between pLBP and control groups

Among the device-determined sleep variables (Table 3), deep sleep was shorter for the pLBP group (1.4±0.4 hr) than the control group (1.7±0.3 hr, p=0.034) (Fig 1). Other variables, including sleep period time and sleep quality and architecture measures were not different between the two groups. Adjusting for age and sex did not impact the results.

Self-reported sleep variables are also presented in Table 3. The pLBP group reported lower sleep quality, less refreshing sleep, more problems with sleep, and greater difficulty falling asleep. The pLBP group also had a higher proportion of participants with a high risk of sleep apnea.

## Associations between self-reported and device-determined sleep

Fig 2 shows the correlations between device-determined and self-reported sleep variables adjusted for age and sex. TST, deep sleep, and REM sleep time were significantly associated with all four self-reported sleep variables (r=0.36 to 0.54, p

**Table 2. Pain experience of participants with persistent low back pain (n=20).**

|  | Mean±SD | Lower quartile | Higher quartile |
|---|---|---|---|
| Tampa Kinesiophobia scale (17–68) | 34.5±5.1 | 32.0 | 38.5 |
| Oswestry Disability Index | 8.2±3.8 | 6.0 | 10.0 |
| Chronic Pain Acceptance |  |  |  |
| Total (0–120) | 78.9±15.2 | 67.0 | 87.0 |
| Engagement (0–66) | 46.2±9.9 | 37.5 | 53.0 |
| Willingness (0–54) | 32.8±7.8 | 27.5 | 39.5 |
| Pain Anxiety Symptom Scale |  |  |  |
| Total (0–100) | 34.7±18.9 | 23.5 | 43.0 |
| Cognitive (0–25) | 11.8±5.4 | 10.0 | 16.0 |
| Escape/Avoidance (0–25) | 10.4±4.6 | 6.5 | 13.5 |
| Fear (0–25) | 7.1±6.2 | 2.0 | 11.0 |
| Physiological anxiety (0–25) | 5.5±6.8 | 0.5 | 10.0 |
| Chronic Low-Back Pain Minimal Dataset |  |  |  |
| Total pain impact (9–50) | 18.6±4.4 | 17.0 | 21.0 |
| Interference (1–5) | 1.9±0.6 | 1.5 | 2.4 |
| Impact functional status (1–5) | 1.6±0.6 | 1.1 | 1.9 |
| Pain intensity (1–10) | 4.6±1.6 | 3.5 | 6.0 |

Score ranges are included for each scale.

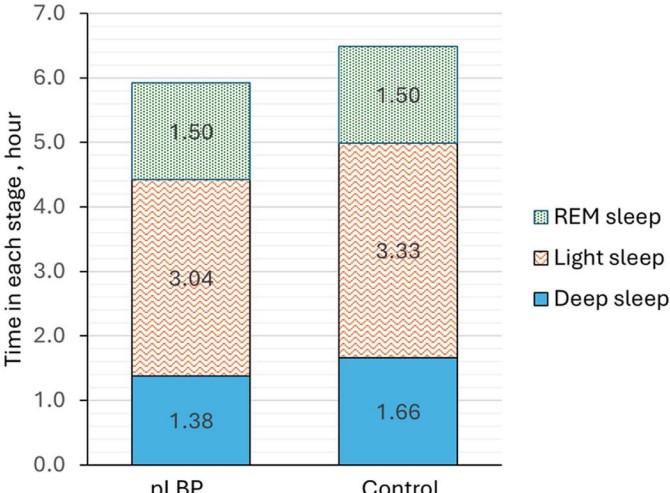

**Fig 1. Time in each sleep stage by group.** The group with persistent low back pain (pLBP) had shorter deep sleep time than the control group (p = 0.034).

range: < 0.001 to 0.037). Sleep period time was associated with self-reported sleep quality, how refreshing sleep was, and problems with sleep. Sleep efficiency was associated with problematic sleep and difficulty falling asleep. Wake after sleep onset was associated with problematic sleep.

We further adjusted for TST in addition to sex and age. Deep sleep time was positively associated (r = 0.35 to 0.42, p = 0.010 to 0.036), while light sleep time was negatively associated (r = -0.35 to -0.38 p = 0.021 to 0.037) with 3 of the four self-reported sleep variables, excluding sleep quality. Other associations mentioned above were no longer significant.

**Associations of device-determined sleep with pain experience and lumbar movement control**

**In the pLBP group.** None of the device-determined sleep variables were significantly associated with fear of movement (Tampa Scale; p values>0.08). Table 4 presents the correlations between sleep variables and scores from other instruments evaluating pain experience. Model 1 was adjusted for age and sex. Model 2 was additionally adjusted for TST. Several associations were significant in both models, including: more wake after sleep onset and shorter deep sleep was each associated with poorer daily function (Oswestry Disability Index); shorter deep sleep was associated with greater pain-related anxiety (PASS); lower sleep efficiency, more wake after sleep onset, and longer latency to REM sleep were associated with greater total pain impact (NIH Dataset); lower sleep efficiency, more wake after sleep onset, and shorter deep sleep were associated with great impact on physical function (NIH Dataset); longer latency to REM sleep was associated with greater pain interference and pain intensity (NIH Dataset).

Additionally, sleep period time, sleep efficiency, wake after sleep onset, REM sleep and light sleep time were associated with CPAQ, PASS, or their subscale scores in model 1, but the associations were lost in model 2. A few other significant associations emerged in Model 2, including: sleep period time and total pain impact and impact on function, deep sleep and pain interference, as well as light sleep and Oswestry Disability Index, total pain impact, pain interference, and impact on function.

**In the entire sample.** In model 1, sleep period time and TST were associated with CES-D and trait anxiety (r range: -0.37 to -0.43, p = 0.008 to 0.025), and deep sleep time was associated with CES-D (r = -0.37, p = 0.026). In model 2, these associations were no longer significant.

**Table 3. Device-determined and self-reported sleep variables.**

| | pLBP N = 20 | Control N = 19 | *P* value |
|---|---|---|---|
| *Device-determined sleep variables* | | | |
| Sleep period time, hr | 6.9 ± 1.2 | 7.7 ± 1.1 | 0.058 |
| Total sleep time, hr | 5.9 ± 1.2 | 6.5 ± 1.1 | 0.138 |
| *Sleep quality* | | | |
| Sleep efficiency, % | 85.1 ± 7.9 | 84.3 ± 8.6 | 0.762 |
| Wake after sleep onset, min | 26.2 ± 17.9 | 33.2 ± 25.5 | 0.329 |
| Latency to persistent sleep, min | 25.6 ± 21.6 | 26.9 ± 18.4 | 0.848 |
| Latency to deep sleep, min | 41.9 ± 41.4 | 29.8 ± 18.4 | 0.244 |
| Latency to REM sleep, min | 102.5 ± 35.3 | 108.7 ± 47.1 | 0.646 |
| *Sleep architecture* | | | |
| Deep sleep time, hr | 1.4 ± 0.4 | 1.7 ± 0.3 | 0.034 |
| % of total sleep as deep sleep, % | 24.1 ± 7.6 | 26.4 ± 6.1 | 0.311 |
| REM sleep time, hr | 1.5 ± 0.5 | 1.5 ± 0.6 | 0.988 |
| % of total sleep as REM sleep, % | 24.3 ± 7.2 | 22.5 ± 5.5 | 0.371 |
| Light sleep time, hr | 3.0 ± 0.7 | 3.3 ± 0.9 | 0.272 |
| % of total sleep as light sleep, % | 51.6 ± 8.5 | 51.1 ± 9.3 | 0.884 |
| *Self-reported sleep variables* | | | |
| Sleep quality (1–5)* | 3.0 ± 1.1 | 3.7 ± 0.8 | 0.025 |
| How refreshing sleep was (1–5)* | 3.0 ± 1.0 | 3.8 ± 0.8 | 0.006 |
| Having problems with sleep (1–5)* | 3.4 ± 1.3 | 4.4 ± 1.0 | 0.008 |
| Difficulty falling asleep (1–5)* | 3.5 ± 1.3 | 4.5 ± 1.0 | 0.009 |
| Berlin questionnaire, n (%) having high risk | 5 (25.0) | 0 (0%) | 0.020 |

*, the question is from the NIH Standards for Research on Chronic Low Back Pain, with higher scores indicating better sleep. Score ranges are included. Mean ± SD are reported.

Also, deep sleep time was associated with the number of positive lumbar control tests (Spearman ρ=-0.35, p = 0.033) in model 1, but not in model 2. Fig 3 is a scatter plot showing the distribution of deep sleep time against the number of positive lumbar control tests.

### Associations of self-reported sleep with pain experience and lumbar movement control

**In the pLBP group.** Individuals who reported more refreshing sleep also reported less pain-related anxiety (PASS anxiety subscale; r = -0.63, p = 0.005, adjusted for age and sex). Individuals who reported more problems with sleep reported less engagement in life activities (CPAQ) (r = 0.48, p = 0.043, adjusted for age and sex).

**In the entire sample.** Self-reported lower sleep quality and less refreshing sleep was associated with more depressive symptoms and state- and trait anxiety (r = -0.42 to -0.48, p < 0.01). Participants who reported more sleep problems had more depressive symptoms, trait anxiety, and a tendency to magnify pain (r = -0.33 to -0.47, p < 0.05). Self-reported lower sleep quality and less refreshing sleep were associated with greater number of positive lumbar control tests (Spearman ρ=-0.33 and -0.36, respectively, p < 0.05).

## Discussion

The significant contribution of our study to the pain and sleep research includes the use of a home sleep test device to ecologically and objectively determine sleep quality and architecture and the exploration of the interplay between sleep

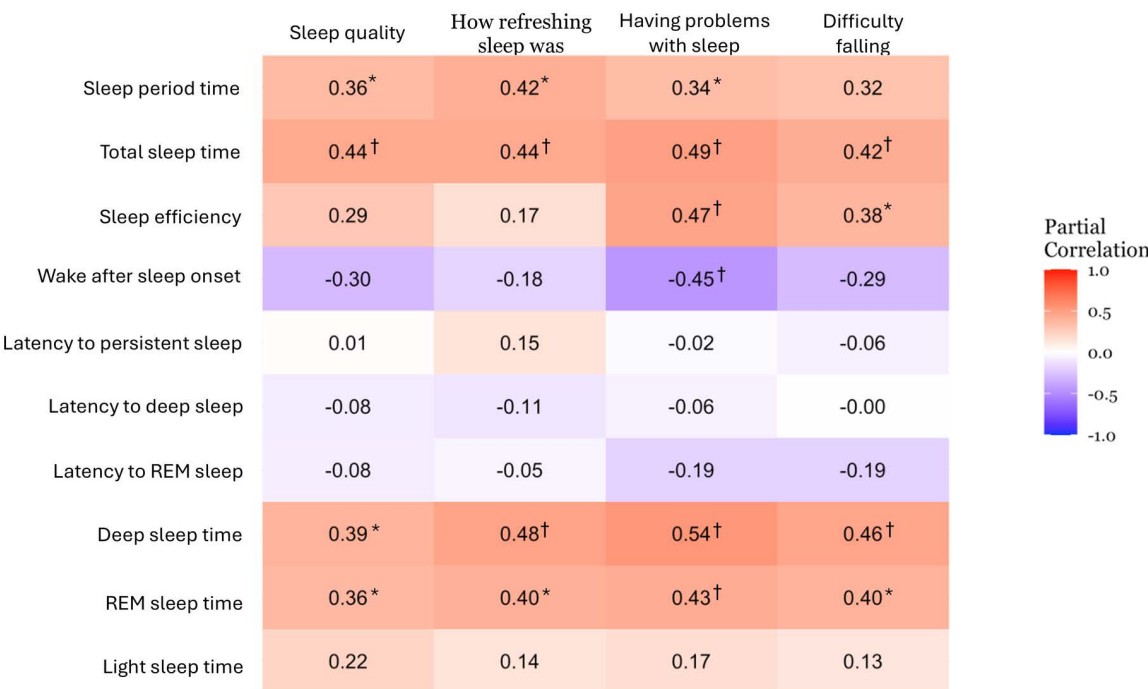

**Fig 2. Associations between device-determined and self-reported sleep variables.** Data are Pearson's correlation coefficients after adjustment for age and sex. *, 0.01 < p < 0.05; †, p < 0.01.

architecture and the self-reported sleep and pain experience in persons with persistent low back pain, for which effective management may have important long-term impact. We found that participants in the pLBP group had shorter deep sleep than the control group. Device-determined deep sleep time and sleep quality measures were associated with multiple aspects of the pain experience and the impact on daily function. On the contrary, self-reported sleep variables were primarily associated with non-pain-specific depression and anxiety symptoms.

Consistent with the literature, participants with pLBP in our study reported worse sleep than the control group. However, among the device-determined sleep quality and architecture variables, the two groups only differed on deep sleep time. Examination of the associations between self-reported and device-determined sleep variables revealed that device-determined deep sleep and TST were associated with all four self-reported sleep questions. Deep sleep was still associated with 3 of the four questions even after adjustment for TST. This finding suggests that deep sleep, among all the sleep stages, was likely the strongest determinant of an individual's self-reported sleep.

Deep sleep time of the pLBP group was shorter than the control group by an average of 18 min. Although there is no consensus regarding the amount of deep sleep needed for maintaining health, one study reported that young people (16–25 years) spent an average of 18.9% of their sleep in the deep stage [28]. This equates to 74 min of deep sleep using the average 6.5-hr TST of the control group, and the 18-min difference in deep sleep between the two groups accounts for a sizable 24% of this amount. The deep sleep stage is characterized by a higher arousal threshold and dominance of slow-wave brain activity [29]. The literature suggests pain potentially interferes with sleep by increasing the internal arousal level [8]. Thus, with the presence of pain, it may be harder to stay in deep sleep and results in short deep sleep in our study. In patients with fibromyalgia, deep sleep has also been reported to be shorter than healthy controls [10]. Classic studies from the 1970's demonstrated that patients with chronic, widespread, musculoskeletal pain showed characteristic abnormalities of non-REM sleep, and on the other hand, selective disruption of slow wave sleep (deep sleep) could

**Table 4. Associations between device-determined sleep variables and pain experiences in participants with chronic low back pain (n = 20).**

| Instrument | Oswestry | CPAQ | | | PASS | | | | | NIH Chronic Low-Back Pain Minimal Dataset | | | |
|---|---|---|---|---|---|---|---|---|---|---|---|---|---|
| Subscale | | Total | Engagement | Willingness | Total | Cognitive | Avoidance | Fear | Anxiety | Total Impact | Interference | Function | Intensity |
| **Sleep period time** | | | | | | | | | | | | | |
| Model 1 | 0.01 | 0.59† | 0.39 | 0.65† | -0.62† | -0.43 | -0.56* | -0.71† | -0.37 | 0.19 | 0.27 | 0.15 | -0.10 |
| Model 2 | 0.31 | -0.17 | -0.34 | 0.16 | -0.23 | -0.30 | -0.31 | -0.25 | 0.07 | 0.50* | 0.39 | 0.50* | 0.02 |
| **Total sleep time** | | | | | | | | | | | | | |
| Model 1 | -0.15 | 0.73† | 0.59† | 0.67† | -0.60† | -0.33 | -0.48* | -0.71† | -0.45 | -0.06 | 0.10 | -0.10 | -0.13 |
| Model 2 | NA | NA | NA | NA | NA | NA | NA | NA | NA | NA | NA | NA | NA |
| **Sleep efficiency** | | | | | | | | | | | | | |
| Model 1 | -0.40 | 0.48* | 0.58* | 0.22 | -0.16 | 0.10 | -0.04 | -0.23 | -0.28 | -0.53* | -0.34 | -0.49* | -0.17 |
| Model 2 | -0.39 | 0.13 | 0.36 | -0.26 | 0.28 | 0.37 | 0.32 | 0.29 | -0.03 | -0.60* | -0.48 | -0.53* | -0.11 |
| **Wake after sleep onset** | | | | | | | | | | | | | |
| Model 1 | 0.57* | -0.34 | -0.55* | 0.03 | 0.23 | 0.04 | 0.09 | 0.20 | 0.36 | 0.56* | 0.32 | 0.77† | -0.14 |
| Model 2 | 0.56* | -0.15 | -0.46 | 0.36 | 0.04 | -0.08 | -0.08 | -0.06 | 0.25 | 0.58* | 0.38 | 0.78† | -0.19 |
| **Latency to persistent sleep** | | | | | | | | | | | | | |
| Model 1 | 0.04 | -0.15 | -0.16 | -0.09 | -0.26 | -0.32 | -0.32 | -0.16 | -0.11 | 0.28 | 0.27 | 0.07 | 0.26 |
| Model 2 | 0.03 | -0.13 | -0.14 | -0.04 | -0.39 | -0.37 | -0.41 | -0.32 | -0.16 | 0.28 | 0.28 | 0.06 | 0.25 |
| **Latency to deep sleep** | | | | | | | | | | | | | |
| Model 1 | 0.34 | -0.06 | -0.18 | 0.11 | -0.16 | -0.23 | -0.26 | -0.08 | -0.01 | 0.40 | 0.41 | 0.27 | 0.06 |
| Model 2 | 0.35 | -0.11 | -0.24 | 0.12 | -0.17 | -0.23 | -0.28 | -0.08 | <0.002 | 0.40 | 0.41 | 0.27 | 0.07 |
| **Latency to REM sleep** | | | | | | | | | | | | | |
| Model 1 | 0.17 | -0.28 | -0.37 | -0.08 | -0.26 | -0.38 | -0.28 | -0.17 | -0.09 | 0.50* | 0.51* | 0.06 | 0.51* |
| Model 2 | 0.17 | -0.38 | -0.44 | -0.08 | -0.35 | -0.41 | -0.34 | -0.27 | -0.12 | 0.50* | 0.51* | 0.06 | 0.51* |
| **Deep sleep time** | | | | | | | | | | | | | |
| Model 1 | -0.65† | 0.35 | 0.45 | 0.12 | -0.43 | -0.11 | -0.18 | -0.43 | -0.59† | -0.45 | -0.40 | -0.48* | 0.11 |
| Model 2 | -0.66† | 0.05 | 0.26 | -0.26 | -0.23 | 0.04 | 0.04 | -0.20 | -0.49* | -0.47 | -0.49* | -0.49* | 0.18 |
| **REM sleep time** | | | | | | | | | | | | | |
| Model 1 | -0.28 | 0.73† | 0.65† | 0.60† | -0.53* | -0.37 | -0.32 | -0.56* | -0.46 | -0.20 | -0.08 | -0.29 | 0.01 |
| Model 2 | -0.28 | 0.28 | 0.33 | 0.05 | -0.01 | -0.17 | 0.23 | 0.18 | -0.15 | -0.31 | -0.34 | -0.40 | 0.26 |
| **Light sleep time** | | | | | | | | | | | | | |
| Model 1 | 0.32 | 0.47* | 0.25 | 0.59† | -0.35 | -0.22 | -0.44 | -0.50* | -0.06 | 0.31 | 0.44 | 0.31 | -0.27 |
| Model 2 | 0.67† | -0.17 | -0.36 | 0.19 | 0.19 | 0.05 | -0.14 | 0.08 | 0.48 | 0.53* | 0.56* | 0.58* | -0.27 |

CPAQ, Chronic Pain Acceptance Questionnaire. PASS, Pain Anxiety Symptom Scale. Pearson's correlation coefficients are presented. In Model 1, age and sex were adjusted. In Model 2, age, sex, and total sleep time were adjusted. NA, not applicable. *, $0.01 < p < 0.05$; †, $p < 0.01$.

reproduce similar abnormalities as well as associated pain symptoms [30,31], supporting a direct link between pain and deep sleep.

Previous meta-analyses also reported less TST, longer light sleep, longer latency in sleep onset, poorer sleep efficiency, and more wake after sleep onset in patients with fibromyalgia [10] and patients with chronic pain and sleep disorders [8] than controls. In our study, however, none of these sleep measures were different between the pLBP and controls. We suspect a primary reason for this is because our participants were younger and those with pLBP had chronic pain for shorter periods (only 30% were longer than five years) than participants in most studies included in the meta-analyses (mean of 6.4 years since diagnosis of fibromyalgia [10] and nine years since experiencing chronic pain [8]). Another reason may be related to the moderate levels of pain and mild disability due to pain for the pLBP group. The

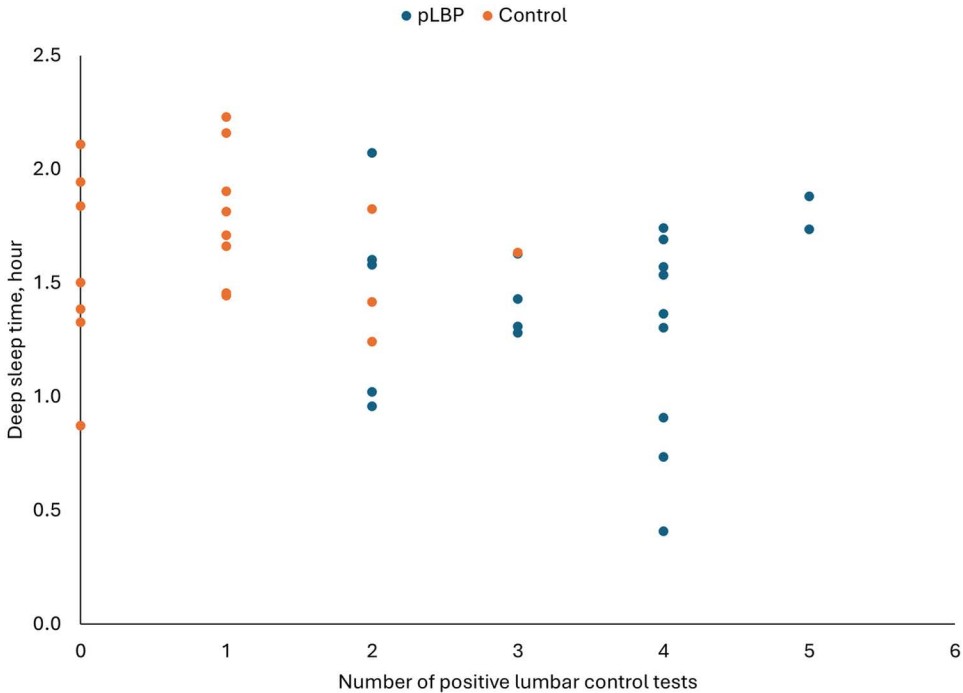

**Fig 3. Scatter plot of deep sleep time and number of positive lumbar control tests. pLBP, persistent low back pain.**

longer an individual has experienced chronic pain and the more severe the pain is, the more likely it will impact sleep and disrupt its continuity. While deep sleep has a restorative function, the longer light sleep found in the meta-analyses [8,10] may be a compensatory response to shortened deep sleep over time. Participants with pLBP in our study may have not yet developed this compensatory change. Also, the association of light sleep with pain experience varied depending on whether TST was adjusted, suggesting light sleep itself may not have an independent association with pain experience.

For the pLBP group, device-determined deep sleep time and sleep quality measures, including sleep efficiency, wake after sleep onset, and latency to REM sleep, were associated with pain-related anxiety and pain impact, whether TST was adjusted or not. Sleep period time, TST, and REM sleep time appear to be associated with similar measures of pain experience; however, adjusting for TST resulted in the loss of most associations of sleep period time and REM sleep time with CPAQ, PASS, and their subscales, suggesting TST mediates the associations. These results align with findings from previous experimental studies that have used REM sleep deprivation protocols. In one study, depriving REM sleep without TST loss did not increase pain even after four consecutive days [32], while another study where TST also reduced, heat pain sensitivity increased [33]. In patients with chronic temporomandibular disorders, TST was positively associated with better pain inhibition, a risk factor for the development of maintenance of persistent pain complaints [34].

Three of the 4 self-reported sleep variables were associated with psychosocial measures of non-pain-specific depression and anxiety. Also, self-reported problems with sleep, but none of the device-determined variables, were significantly associated with pain catastrophizing. Consistent with the literature [35,36], TST (device-determined) was associated with depression and trait anxiety; however, other device-determined sleep variables were not associated with these psychosocial measures after TST was adjusted. On the other hand, few significant associations were found for self-reported sleep with pain-specific experiences. Self-reported sleep can be biased by factors other than sleep per se, such as stress, mood, and recall bias. Typically, individuals are asked to provide an overall perception for the past few days to weeks, specifically, seven days in our study. Objective sleep measures are usually conducted on specific nights, and three nights

in our study. A review comparing objective measures of sleep and self-reported sleep quality found that TST and sleep efficiency, but not other objective sleep variables, were reliably predictive of self-reported sleep quality [37]. Our results demonstrate differences in the implications of self-reported and device-determined sleep, in that self-reported sleep is more linked to perceived general depressive symptoms and anxiety.

The literature and our study also support a role of sleep in motor control. Acute and chronic sleep deprivation impaired performance in the sensorimotor synchronization gait control in young adults [38]. A daytime nap promotes motor learning and motor skill consolidation [39]. In our study, shorter deep sleep time (TST not adjusted), and lower self-reported sleep quality and less refreshing sleep were associated with poorer lumbar movement control. These results generally support that sleep has a role in motor control. However, the roles of different stages of sleep in relation to motor control and motor learning are complex, and their roles may be different in memory consolidation and motor learning [40–42]. The findings of less deep sleep of participants with pLBP group in our study may have implications for learning new motor patterns due to ongoing back pain.

There were several strengths of this study. Although this ambulatory device is not the gold-standard polysomnography, it determines sleep architecture and quality based on EEG and therefore provided greater ecological insight into the relationship between sleep and pain that otherwise would not be possible. This device was used at home, a familiar environment for participants, and is more reflective of habitual sleep. Self-reported sleep was also collected, allowing the comparison between the two types of sleep measures in association with pain experience. Another strength of this study was the availability of psychosocial information from various instruments, including pain-related and information not specific to pain. Furthermore, the availability of movement control tests allowed us to examine the association of sleep with movement impairment.

However, our sample size was not large, and this study examines cross-sectional associations, which does not allow any inference of causality. Also, the participants were young adults, their pain intensity was generally light to moderate, and the duration of pain was less than five years for 70% of them. Therefore, findings may not be generalizable to populations of patients with older age, more severe pain, or longer duration of pain. However, these individuals are an important population to study because effective management may have greater chance to improve or even reverse symptoms, and thus the long-term impact of treatment will be greater for these individuals and society. Another limitation is that sleep and movement control tests were not always on the same days. The tests were completed within 10 days of each other and the pLBP participants were not currently engaged in rehabilitation, which could help reduce the influence of day-to-day change in movement control on results. In addition, the raters for the movement control test were not blinded to group and this may have influenced our findings. Medications could impact sleep; however, only three of the 20 participants used medications to control pain.

In conclusion, our study showed that young adults with pLBP had shorter deep sleep than controls. Deep sleep time and certain sleep quality measures were associated with several psychosocial measures of pain-related anxiety and impact of pain on function and daily activities, independent of TST. Future studies are needed to examine whether lengthening deep sleep, TST, or improving sleep quality can improve pain-related psychosocial experience in individuals with pLBP.

## Author contributions

**Conceptualization:** Xuewen Wang, Jennifer M. C. Vendemia, Sheri P. Silfies.

**Data curation:** Xuewen Wang, Jennifer M. C. Vendemia, Erin E. Kishman, John R. Gilliam, Sheri P. Silfies.

**Formal analysis:** Xuewen Wang.

**Funding acquisition:** Xuewen Wang, Jennifer M. C. Vendemia, Sheri P. Silfies.

**Investigation:** Erin E. Kishman, John R. Gilliam, Sheri P. Silfies.

**Methodology:** Xuewen Wang, Jennifer M. C. Vendemia, Alexandria M. Reynolds, Sheri P. Silfies.

**Project administration:** Xuewen Wang, Jennifer M. C. Vendemia, Erin E. Kishman, John R. Gilliam, Sheri P. Silfies.

**Resources:** Xuewen Wang, Jennifer M. C. Vendemia.

**Supervision:** Xuewen Wang, Jennifer M. C. Vendemia, Sheri P. Silfies.

**Visualization:** Xuewen Wang.

**Writing – original draft:** Xuewen Wang.

**Writing – review & editing:** Jennifer M. C. Vendemia, Erin E. Kishman, John R. Gilliam, Alexandria M. Reynolds, Sheri P. Silfies.

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
