## [Decision Letter · Decision Letter 0]

20 Jan 2025

PONE-D-24-49852Sleep Architecture and Quality and Pain Experience in Individuals with Persistent Low Back Pain and Asymptomatic ControlsPLOS ONE

Dear Dr. Wang,

Thank you for submitting your manuscript to PLOS ONE. After careful consideration, we feel that it has merit but does not fully meet PLOS ONE’s publication criteria as it currently stands. Therefore, we invite you to submit a revised version of the manuscript that addresses the points raised during the review process.

We look forward to receiving your revised manuscript.

Kind regards,

Peng Zhong, Ph.D.

Academic Editor

PLOS ONE

“National Institutes of Health under Award Number R01HD095959 to University of South Carolina, PI: Vendemia and Silfies. The University of South Carolina ASPIRE 80004373, PI: Wang.”

Reviewers' comments:

Reviewer's Responses to Questions

**Comments to the Author**

1. Is the manuscript technically sound, and do the data support the conclusions?

Reviewer #1: Yes

Reviewer #2: Partly

2. Has the statistical analysis been performed appropriately and rigorously? 

Reviewer #1: Yes

Reviewer #2: Yes

3. Have the authors made all data underlying the findings in their manuscript fully available?

Reviewer #1: Yes

Reviewer #2: Yes

4. Is the manuscript presented in an intelligible fashion and written in standard English?

Reviewer #1: Yes

Reviewer #2: Yes

5. Review Comments to the Author

Reviewer #1: The article titled "Sleep Architecture and Quality and Pain Experience in Individuals with Persistent Low Back Pain and Asymptomatic Controls" addresses a significant yet underexplored topic—the interplay between chronic low back pain (pLBP) and sleep architecture. The study’s nuanced approach in using both device-determined and self-reported sleep measures to investigate this relationship offers a deeper understanding, particularly in young adults experiencing persistent, albeit moderate, pain.

The research hinges on a foundational premise: the bidirectional nature of pain and sleep disturbances. Chronic pain is known to disrupt sleep quality, while inadequate sleep heightens pain perception. What sets this study apart is its focus on specific sleep architecture elements, assessed through an ambulatory EEG device, allowing participants to record sleep patterns in their natural home environment. This method enhances ecological validity and provides richer insights compared to lab-based sleep studies, which often miss the subtleties of habitual sleep.

Key findings reveal that participants with pLBP had significantly shorter durations of deep sleep compared to controls, though other sleep measures, such as total sleep time and REM sleep, were not markedly different. The reduction in deep sleep—a stage critical for restorative processes and pain modulation—is particularly telling. While the 18-minute average difference might seem minor in absolute terms, its potential physiological impact is notable, especially given deep sleep's role in overall health and well-being. Interestingly, the study also found that deep sleep correlated strongly with participants’ self-rated sleep quality, emphasizing its subjective and objective importance.

The article goes beyond describing differences, delving into correlations between sleep variables and psychosocial factors. For the pLBP group, poor sleep efficiency, longer wake periods, and reduced deep sleep were associated with heightened pain-related anxiety and greater daily functional limitations. These connections underline the complex feedback loop where disrupted sleep exacerbates pain and vice versa. However, the researchers also identified that self-reported sleep disturbances were more closely linked to general emotional states, such as depression and anxiety, than to pain-specific experiences. This distinction between device-determined and self-reported sleep metrics is an insightful contribution, shedding light on the subjective biases inherent in sleep evaluations.

The study’s focus on movement control adds another layer of relevance, particularly in understanding how sleep disturbances might impair motor function. The finding that reduced deep sleep correlates with poorer lumbar control underscores the need for further research into sleep’s role in neuromuscular rehabilitation. This association, while intriguing, raises questions about causality and whether improving sleep quality could directly enhance movement control in individuals with pLBP.

While the study’s strengths are evident, including its methodological rigor and comprehensive psychosocial assessments, its limitations are equally clear. The sample size is relatively small, and the participants’ young age and moderate pain levels may limit the generalizability of findings to older populations or those with more severe, long-standing pain. Additionally, the cross-sectional design precludes causal inferences, leaving open the question of whether improving sleep would mitigate pain symptoms or vice versa.

In summary, this study makes a compelling case for prioritizing deep sleep in the management of chronic pain conditions. Its findings bridge gaps in existing literature by linking sleep architecture to pain and functional outcomes, particularly in young adults with pLBP. Future research, ideally with larger and more diverse samples, could explore interventions targeting deep sleep improvement—such as cognitive-behavioral therapy for insomnia or non-pharmacological sleep aids—to assess their efficacy in reducing pain and enhancing quality of life. This paper is a valuable step forward, blending clinical relevance with methodological sophistication, and it prompts critical discussions about the often-overlooked role of sleep in pain management.

Reviewer #2: The authors conducted an observational study on the relationships between sleep architecture and subjective sleep quality and pain and psychosocial factors in participants with and without chronic low back pain. This is an intriguing study, providing additional insight into how sleep architecture differs between people with and without chronic low back pain. Well done to the research team!

Methodology

There are a few important methodological aspects missing, which I think need to be to include.

1. Blinding strategies have not been reported. Specifically, please can the authors clarify whether participants were or were not blinded to group allocation and/or research questions/hypotheses. If there were strategies to blind participants, please can the authors explain said strategies and report on blinding assessments to determine whether blinding of participants was or was not successful.

2. Also related to blinding, were (1) assessors and (2) data analysts blinded to group allocation? Specifically, it would have been important for assessors to have been blinded to group allocation during the lumbar movement control evaluation, and it would have been important for the data analysts to have been blinded to group allocation when conducting and interpreting the data analysis. Please can the authors provide clarity on this.

3. The authors have not provided any information on sample size calculations, as such it is unclear whether the study is adequately powered. Please can the authors provide clarity on this.

4. Please can the authors briefly discuss the validity and reliability of the lumbar movement control evaluation as an accurate measure of “movement control of the lumbar spine”.

5. Similarly, please can the authors briefly discuss the validity and reliability of the Zmachine® Insight+. It is unclear what exactly ‘substantial agreement’ means in reference to the phase in line 134 “which substantially agrees with in-lab polysomnography”.

Results

1. Table 1: with reference to race/ethnicity, can the authors rather describe what the race/ethnicity was than what it was not. Saying “non-white/non-Hispanic” is non-specific. It is far more accurate to describe something by what it is rather than what is it not. Additionally, please clarify if this was participant-reported race/ethnicity or some other method to determine/assess race/ethnicity.

2. Lines 208 – 210: please included numerator and denominators in brackets next to percentages, otherwise percentages appear inflated given the small sample size.

3. Line 213, given medication can influence sleep, please can the authors include a detailed description of the medication and injections used.

4. Please include p values in lines 244, 254, and 283 when reporting r statistics.

5. Table 4: I think plots would provide more insight than r values. I recommend the authors include a correlation matrix of the variables presented in table 4 so that the reader can more clearly visualise the relationship between each of the variables. Ditto for table 5.

6. I recommend the authors improve the quality of figures 1 and 2; in their current state they are very pixelated.

Discussion

1. I don’t think it is accurate to state the chronic low back pain is an “understudied chronic pain population”. I recommend the authors provide clarity on why they argue this is an understudied population.

2. Can the authors discuss a bit further why there were differences in finding between the objective assessment of sleep architecture and subjective sleep quality assessment by tapping into the literature on subjective vs objective measures of sleep quality.

3. I think the discussion could benefit from a discussion of proposed mechanisms for why in this study deep sleep time was less in people with pLBP than controls without pain, with reference to the bidirectional relationship between sleep and pain, referred to in the introduction.

4. Line 373, I disagree that Parkinson’s is a good example to use here. It is far more complicated than a straightforward motor disorder. Eg. degeneration of dopaminergic neurons seen in Parkinson’s influence sleep regulation.

5. Line 374, specifically what about REM sleep “emerged as a biomarker …”

6. Limitations: please can the authors comment on how medication use may influence sleep architecture.

6. PLOS authors have the option to publish the peer review history of their article (what does this mean? ). If published, this will include your full peer review and any attached files.

**Do you want your identity to be public for this peer review?** For information about this choice, including consent withdrawal, please see our Privacy Policy .

Reviewer #1: **Yes: ** Denis Banchenko

Reviewer #2: No

---

## [Author Response · Author response to Decision Letter 1]

5 Mar 2025

We thank the reviewers for their thorough review and their overall positive view of our manuscript. Revisions, as detailed below, have been made according to the reviewer’s comments. The revisions are tracked in the manuscript and the line numbers included in this document refer to those in this revised manuscript. We believe these revisions have improved the quality of this manuscript. Please see below our point-by-point responses to reviewers.

Reviewer 1:

The article titled "Sleep Architecture and Quality and Pain Experience in Individuals with Persistent Low Back Pain and Asymptomatic Controls" addresses a significant yet underexplored topic—the interplay between chronic low back pain (pLBP) and sleep architecture. The study’s nuanced approach in using both device-determined and self-reported sleep measures to investigate this relationship offers a deeper understanding, particularly in young adults experiencing persistent, albeit moderate, pain.

The research hinges on a foundational premise: the bidirectional nature of pain and sleep disturbances. Chronic pain is known to disrupt sleep quality, while inadequate sleep heightens pain perception. What sets this study apart is its focus on specific sleep architecture elements, assessed through an ambulatory EEG device, allowing participants to record sleep patterns in their natural home environment. This method enhances ecological validity and provides richer insights compared to lab-based sleep studies, which often miss the subtleties of habitual sleep.

Key findings reveal that participants with pLBP had significantly shorter durations of deep sleep compared to controls, though other sleep measures, such as total sleep time and REM sleep, were not markedly different. The reduction in deep sleep—a stage critical for restorative processes and pain modulation—is particularly telling. While the 18-minute average difference might seem minor in absolute terms, its potential physiological impact is notable, especially given deep sleep's role in overall health and well-being. Interestingly, the study also found that deep sleep correlated strongly with participants’ self-rated sleep quality, emphasizing its subjective and objective importance.

The article goes beyond describing differences, delving into correlations between sleep variables and psychosocial factors. For the pLBP group, poor sleep efficiency, longer wake periods, and reduced deep sleep were associated with heightened pain-related anxiety and greater daily functional limitations. These connections underline the complex feedback loop where disrupted sleep exacerbates pain and vice versa. However, the researchers also identified that self-reported sleep disturbances were more closely linked to general emotional states, such as depression and anxiety, than to pain-specific experiences. This distinction between device-determined and self-reported sleep metrics is an insightful contribution, shedding light on the subjective biases inherent in sleep evaluations.

The study’s focus on movement control adds another layer of relevance, particularly in understanding how sleep disturbances might impair motor function. The finding that reduced deep sleep correlates with poorer lumbar control underscores the need for further research into sleep’s role in neuromuscular rehabilitation. This association, while intriguing, raises questions about causality and whether improving sleep quality could directly enhance movement control in individuals with pLBP.

While the study’s strengths are evident, including its methodological rigor and comprehensive psychosocial assessments, its limitations are equally clear. The sample size is relatively small, and the participants’ young age and moderate pain levels may limit the generalizability of findings to older populations or those with more severe, long-standing pain. Additionally, the cross-sectional design precludes causal inferences, leaving open the question of whether improving sleep would mitigate pain symptoms or vice versa.

In summary, this study makes a compelling case for prioritizing deep sleep in the management of chronic pain conditions. Its findings bridge gaps in existing literature by linking sleep architecture to pain and functional outcomes, particularly in young adults with pLBP. Future research, ideally with larger and more diverse samples, could explore interventions targeting deep sleep improvement—such as cognitive-behavioral therapy for insomnia or non-pharmacological sleep aids—to assess their efficacy in reducing pain and enhancing quality of life. This paper is a valuable step forward, blending clinical relevance with methodological sophistication, and it prompts critical discussions about the often-overlooked role of sleep in pain management.

Author response: We thank the reviewer for the thorough review and accurate summary of our findings, and we appreciate the assessment and praise of the value of our work.

Reviewer #2: The authors conducted an observational study on the relationships between sleep architecture and subjective sleep quality and pain and psychosocial factors in participants with and without chronic low back pain. This is an intriguing study, providing additional insight into how sleep architecture differs between people with and without chronic low back pain. Well done to the research team!

Methodology

There are a few important methodological aspects missing, which I think need to be to include.

1. Blinding strategies have not been reported. Specifically, please can the authors clarify whether participants were or were not blinded to group allocation and/or research questions/hypotheses. If there were strategies to blind participants, please can the authors explain said strategies and report on blinding assessments to determine whether blinding of participants was or was not successful.

Author response: Participants were grouped based on whether they had persistent pain and therefore they were not blinded to group allocation. They were told the research was to examine associations between sleep and pain by examining sleep in individuals with and without chronic low back pain and they were not informed of the research hypotheses.

2. Also related to blinding, were (1) assessors and (2) data analysts blinded to group allocation? Specifically, it would have been important for assessors to have been blinded to group allocation during the lumbar movement control evaluation, and it would have been important for the data analysts to have been blinded to group allocation when conducting and interpreting the data analysis. Please can the authors provide clarity on this.

Author response: The assessors of the lumbar movement control tests were all licensed physical therapists trained in the tests and their dichotomous determination of positive or negative rating. Each test had specific criteria that was observed or measured related to the participants’ ability to control the relationship between the lumbar spine and pelvis. Assessors were not blinded to group allocation. We added this as a limitation (lines 431-432).

All survey assessments were conducted using the REDCap platform, where participants answered the questions directly online. Objective sleep was determined and calculated automatically by the software. The data would not be influenced by blinding.

3. The authors have not provided any information on sample size calculations, as such it is unclear whether the study is adequately powered. Please can the authors provide clarity on this.

Author response: Inadequate information was available to conduct a full power analysis, and this study was ancillary to an ongoing study. Therefore, the sample size was primarily determined given the availability of resources and the feasibility of adding this component to the parent study. It was estimated that a sample size of 25 in each group would provide 80% power to detect a large effect size (Cohen’s d=0.81) for comparing means between the two groups, using a significance level of 0.05 with a two-tailed t test. This was added to the manuscript (lines 187-191).

4. Please can the authors briefly discuss the validity and reliability of the lumbar movement control evaluation as an accurate measure of “movement control of the lumbar spine”.

Author response: Prior work has established substantial intra and inter-rater reliability of these six tests (kappa > 0.60) in both novice and experienced clinicians (Luomajoki et al, 2007). A follow-up study by Luomajoki et al, 2008 demonstrated a significant difference in the number of positive tests between patients with low back pain (2.21; 95%CI 1.94-2.48) and subjects without back pain (0.75 ;95%CI 0.55-0.95) regarding their ability to actively control the movements of the low back during the six tests (0-6 points). The effect size between patients with low back pain and healthy controls in movement control was large (d =1.18; p < 0.001). We added some of this information to the manuscript (lines 177-178, 183-185).

Luomajoki H, Kool J, de Bruin ED, Airaksinen O. Reliability of movement control tests in the lumbar spine. BMC Musculoskelet Disord. 2007 Sep 12;8:90. doi: 10.1186/1471-2474-8-90. PMID: 17850669; PMCID: PMC2164955.

Luomajoki H, Kool J, de Bruin ED, Airaksinen O. Movement control tests of the low back; evaluation of the difference between patients with low back pain and healthy controls. BMC Musculoskelet Disord. 2008 Dec 24;9:170. doi: 10.1186/1471-2474-9-170. PMID: 19108735; PMCID: PMC2635372.

5. Similarly, please can the authors briefly discuss the validity and reliability of the Zmachine® Insight+. It is unclear what exactly ‘substantial agreement’ means in reference to the phase in line 134 “which substantially agrees with in-lab polysomnography”.

Author response: In the referenced study, sensitivities and positive predictive values for sleep stages determined by the Zmachine® Insight algorithm, compared to the polysomnography using a consensus of expert scorers, were between 0.72 and 0.85. The overall kappa agreement was 0.72, indicative of substantial agreement. We have added this information to the manuscript (lines 133-136).

Results

1. Table 1: with reference to race/ethnicity, can the authors rather describe what the race/ethnicity was than what it was not. Saying “non-white/non-Hispanic” is non-specific. It is far more accurate to describe something by what it is rather than what is it not. Additionally, please clarify if this was participant-reported race/ethnicity or some other method to determine/assess race/ethnicity.

Author response: The race/ethnicity in Table 1 has been expanded to include all racial/ethnic composition. Participants reported their race/ethnicity, which was clarified (lines173-174).

2. Lines 208 – 210: please included numerator and denominators in brackets next to percentages, otherwise percentages appear inflated given the small sample size.

Author response: The numerators and denominators have been added (lines 220-221).

3. Line 213, given medication can influence sleep, please can the authors include a detailed description of the medication and injections used.

Author response: We did not collect detailed information on medication use outside the current use or not for pain control. The steroid or facet joint injections did not coincide with our testing sessions. More descriptions were added (line 224).

4. Please include p values in lines 244, 254, and 283 when reporting r statistics.

Author response: The p values have been added (lines 255, 268-269, and 297-298).

5. Table 4: I think plots would provide more insight than r values. I recommend the authors include a correlation matrix of the variables presented in table 4 so that the reader can more clearly visualise the relationship between each of the variables. Ditto for table 5.

Author response: Table 4 has been revised to a figure (Figure 2) using a color scheme to indicate strength of correlations. Table 5 includes correlations from two models. We wanted to clearly and effectively display the different models side by side and by subdomains within each instrument, and thus we did not change it.

6. I recommend the authors improve the quality of figures 1 and 2; in their current state they are very pixelated.

Author response: Thank you for the recommendation. The resolution of the figures has been increased.

Discussion

1. I don’t think it is accurate to state the chronic low back pain is an “understudied chronic pain population”. I recommend the authors provide clarity on why they argue this is an understudied population.

Author response: We has clarified our open sentence to the following: “A significant contribution of our study is the use of a home sleep test device to ecologically and objectively determine sleep quality and architecture and the exploration of the interplay between the self-reported pain experience and sleep architecture in persons with persistent low back pain, for which effective management may have important long-term impact.” Lines 322-325.

2. Can the authors discuss a bit further why there were differences in finding between the objective assessment of sleep architecture and subjective sleep quality assessment by tapping into the literature on subjective vs objective measures of sleep quality.

Author response: We added a few sentences (lines 387-393) as below:

“Self-reported sleep can be biased by factors other than sleep per se, such as stress, mood, and recall bias. Typically, individuals are asked to provide an overall perception for the past few days to weeks, specifically, seven days in our study. Objective sleep measures are usually conducted on specific nights, and three nights in our study. A review comparing objective measures of sleep and self-reported sleep quality found that TST and sleep efficiency, but not other objective sleep variables, were reliably predictive of self-reported sleep quality.”

3. I think the discussion could benefit from a discussion of proposed mechanisms for why in this study deep sleep time was less in people with pLBP than controls without pain, with reference to the bidirectional relationship between sleep and pain, referred to in the introduction.

Author response: Thank you for the suggestion. We expanded the discussion on deep sleep in the third paragraph of Discussion (lines 344-347, 350-352).

4. Line 373, I disagree that Parkinson’s is a good example to use here. It is far more complicated than a straightforward motor disorder. Eg. degeneration of dopaminergic neurons seen in Parkinson’s influence sleep regulation.

Author response: We appreciate this point and have thus removed the sentence.

5. Line 374, specifically what about REM sleep “emerged as a biomarker …”

Author response: We removed this sentence as it is related to movement disorders. We added more discussion related to sleep and motor control and motor learning, which are more relevant to our study (lines 400-401, 404-408).

6. Limitations: please can the authors comment on how medication use may influence sleep architecture.

Author response: We added a sentence to the paragraph on limitations. Medications could impact sleep, but only 3 of the 20 participants used medications to control pain (lines 433-434).

---

## [Decision Letter · Decision Letter 1]

16 Mar 2025

Sleep Architecture and Quality and Pain Experience in Individuals with Persistent Low Back Pain and Asymptomatic Controls

PONE-D-24-49852R1

Dear Dr. Wang,

We’re pleased to inform you that your manuscript has been judged scientifically suitable for publication and will be formally accepted for publication once it meets all outstanding technical requirements.

Kind regards,

Peng Zhong, Ph.D.

Academic Editor

PLOS ONE

Additional Editor Comments (optional):

Reviewers' comments:

Reviewer's Responses to Questions

**Comments to the Author**

1. If the authors have adequately addressed your comments raised in a previous round of review and you feel that this manuscript is now acceptable for publication, you may indicate that here to bypass the “Comments to the Author” section, enter your conflict of interest statement in the “Confidential to Editor” section, and submit your "Accept" recommendation.

Reviewer #2: All comments have been addressed

2. Is the manuscript technically sound, and do the data support the conclusions?

Reviewer #2: Yes

3. Has the statistical analysis been performed appropriately and rigorously? 

Reviewer #2: Yes

4. Have the authors made all data underlying the findings in their manuscript fully available?

Reviewer #2: No

5. Is the manuscript presented in an intelligible fashion and written in standard English?

Reviewer #2: Yes

6. Review Comments to the Author

Reviewer #2: (No Response)

7. PLOS authors have the option to publish the peer review history of their article (what does this mean? ). If published, this will include your full peer review and any attached files.

**Do you want your identity to be public for this peer review?** For information about this choice, including consent withdrawal, please see our Privacy Policy .

Reviewer #2: No

---

## [Editor Report · Acceptance letter]

PONE-D-24-49852R1

PLOS ONE

Dear Dr. Wang,

I'm pleased to inform you that your manuscript has been deemed suitable for publication in PLOS ONE. Congratulations! Your manuscript is now being handed over to our production team.

Kind regards,

on behalf of

Dr. Peng Zhong

Academic Editor

PLOS ONE